# Change in Rumination Behavior Parameters around Calving in Cows with Subclinical Ketosis Diagnosed during 30 Days after Calving

**DOI:** 10.3390/ani13040595

**Published:** 2023-02-08

**Authors:** Ramūnas Antanaitis, Vida Juozaitienė, Karina Džermeikaitė, Dovilė Bačėninaitė, Greta Šertvytytė, Eduardas Danyla, Arūnas Rutkauskas, Lorenzo Viora, Walter Baumgartner

**Affiliations:** 1Large Animal Clinic, Veterinary Academy, Lithuanian University of Health Sciences, Tilžės 18, LT-47181 Kaunas, Lithuania; 2Department of Biology, Faculty of Natural Sciences, Vytautas Magnus University, K. Donelaičio 58, LT-44248 Kaunas, Lithuania; 3Scottish Centre for Production Animal Health and Food Safety, School of Biodiversity, One Health and Veterinary Medicine, University of Glasgow, Glasgow G12 8QQ, UK; 4University Clinic for Ruminants, University of Veterinary Medicine, Veterinaerplatz 1, A-1210 Vienna, Austria

**Keywords:** precision dairy farming, prognosis of diseases after calving, sensors, rumination

## Abstract

**Simple Summary:**

The aim of our study was to evaluate whether there are changes in rumination behavior parameters around calving in cows with subclinical ketosis (SCK), diagnosed within the first 30 days of calving, compared with cows without SCK. In the cows with a higher risk of having SCK diagnosed after calving, we found changes in rumination behavior parameters before calving, on the calving day, and after calving. Based on this, we can conclude that by tracking changes in rumination behavior parameters registered with RuniWatch sensors (such as rumination time, eating time, drinking time, drinking gulps, bolus, chews per minute, chews per bolus, downtime, maximal temperature, and activity change) before, during, and after calving, we can identify cows with a higher risk of SCK.

**Abstract:**

We hypothesized that cows with SCK (blood BHB over >1.2 mmol/L) diagnosed within the first 30 days of calving can be predicted by changes in rumination and activity behavioral parameters in the period before calving and indeed subsequently. A total of 45 cows were randomly selected from 60 dry cows from at least 40 days before calving. All the cows were fitted with RuniWatch sensors monitoring both intake behaviors (faceband) and general movement and activity behavior (pedometer) (RWS-ITIN + HOCH, Switzerland). Following an adaptation period of 10 days, rumination, eating, and activity parameters were monitored for 30 days before calving and 30 days after calving. Considering the design of the study, we divided the data of cows into three stages for statistical evaluation: (1) the last thirty days before calving (from day −30 to −1 of the study); (2) day of calving; and (3) the first thirty days after calving (from day 1 to 30 of the study). We found that before calving, those cows with a higher risk of having SCK diagnosed after calving had lower rumination time, eating time, drinking gulps, bolus, chews per min, chews per bolus, downtime, maximal temperature, and activity change. On the calving day, in cows with higher risk of SCK after calving, we found lower rumination time, eating time, chews per min, chews per bolus, uptime, downtime, minimal temperature, other chews, eating chews, drinking time, drinking gulps, activity, average temperature, maximal temperature, activity change, rumination chews, and eating chews. After calving in cows with SCK, we found lower rumination time, eating time 1, eating time 2, bolus, chews per bolus, uptime, downtime, minimal temperature, maximal temperature, rumination chews, and eating chews. Moreover, after calving we found higher drinking gulps, drinking time, activity, activity change, average temperature, other chews, and eating chews in cows with SCK. From a practical point of view, we recommend that by tracking changes in rumination and activity behavior parameters registered with RuniWatch sensors (such as rumination time, eating time, drinking time, drinking gulps, bolus, chews per minute, chews per bolus, downtime, maximal temperature, and activity change) before, during, and after calving, we can identify cows with a higher risk of SCK in the herd.

## 1. Introduction

Automation, in order to reduce (physical) labor and other expenses, is increasingly being applied in dairy farming [1]. This trend is influenced in part by the economic realities of rising labor expenses relative to capital costs. Automated technologies enable dairy farmers to manage larger herds with less labor, implying that the use of automated systems is, at least in part, driving the trend of growing herd sizes [2]. A significant proportion of dairy cows suffer from health problems in the early post-partum period, which has a severe impact on their health, welfare, and performance [3,4]. 

With the availability of automatic monitoring of eating and rumination behavior, the interaction of disease, feed intake and management in the transition period has been highlighted [5,6]. For example, it was found that changes in DMI and time spent feeding in the days before calving indicated cows with diseases post-partum, such as metritis or subclinical ketosis (SCK) [7,8]. Overall, calving disorders are quite common and increase medicine usage, veterinary costs, culling rate, and production and reproduction losses. Farmers require timely and realistic information in order to make immediate decisions that will assist and limit the occurrence of calving illness [9].

Sensor data can be used alone or in combination with established health-monitoring techniques to detect cows with health problems [2]. The RumiWatch system involving eating and rumination behavior via a nose band alongside activity monitoring by pedometer has been validated and widely applied. The rumination behavior can be detected by tracking pressure changes and masticatory intervals of the temporal fossa or noseband while chewing [2,10]. The high to extremely high correlations between direct observations and sensor data demonstrate that the RumiWatch noseband sensor was effectively developed and validated as a scientific monitoring device for the automated detection of rumination and eating behaviors in stable-fed dairy cows [10]. Subclinical ketosis was discovered to be associated with altered rumination and activity behavior in cows. 

The concentration of beta-hydroxybutyrate (BHB) is often used to define SCK. Previous studies reported that cows with blood BHB concentrations over >1.4 mmol/L at the start of lactation may require extra treatment to reduce their risk of adverse outcomes in early lactation [4]. Early detection of SCK is essential to enable timely management changes and health interventions [11]. Based on past findings, we found that subclinical ketosis has a substantial relationship with rumination and activity behaviors. Cows with SCK were more likely to switch between activities (ruminating, eating, and drinking) more frequently with shorter ruminating times, reduced number of chews, as well as shorter drinking times, chews per minute, boluses (total amount of gulps taken while drinking), and chews per bolus. Moreover, a decrease in rumination time was associated with a significant rise in the likelihood of SCK risk. Based on our findings, we suggest that changes in rumination prior to the appearance of clinical indications of SCK could be used to aid in the early detection of the disease [12,13]. 

To our knowledge, limited research has been conducted to assess the relationship between SCK and the battery of behavior changes identified by the RuniWatch system during late transition and the calving period in cows, such as rumination time, eating time, and drinking time, in the transition period including around the time of calving. The aim of our study was to evaluate whether there are changes in rumination behavior parameters in cows with SCK diagnosed within the first 30 days of calving compared with cows without SCK. We hypothesized that alterations in rumination and activity patterns of cows during the 30 days prior to calving could predict a subclinical ketosis diagnosis within the first 30 days post-calving.

## 2. Materials and Methods

### 2.1. Animal and Farm Enrolment

The provisions of the Lithuanian Law on Animal Welfare and Protection were followed in the conduct of this investigation. PK016965 is the study’s approval number. From June 2022 to October 2022, the experiment was conducted at the one Lithuanian dairy farm with 630 milking Holstein cows (55.792368° N, 24.017499° E). The average milk production of the farm was 35 kg/day with an average feed intake of 23.2 kg DM/day, milk fat of 4.32 (±0.15), milk protein of 3.35 (±0.25), milk somatic cell count of 182,000/mL (±0.55), and milk urea nitrogen of 27 (±7). The cows were housed all year round in a free stall barn and milked with a Delaval milking parlor. The average number of calving cows was 60 per month. Cows were fed in accordance with NRC recommendations (Table 1 and Table 2).

### 2.2. Research Design

A total of 45 cows were randomly selected from 60 dry cows 40 days before calving, and RumiWatch sensors (RWS) (RumiWatch System, Itin+Hoch GmbH, Liestal, Switzerland) were applied. Every day at 09:00, these cows were checked according to the clinical examination protocol and blood and milk samples were taken for BHB and milk composition identification. All cows (n = 45) were known to have calved easily in the previous lactation (1–3). The bulls selected for the cows in the study had high breeding value for ease calving.

### 2.3. Study Groups

The SCK group (TG) (n = 20) contained cows with at least one blood beta-hydroxybutyrate acetate (BHB) value of 1.2 mmol/L or above and the milk fat/protein ratio (F/P) was greater than 1.2 during the 30-day post-partum period. Those cows showed no clinical signs of diseases after calving, such as metritis, lameness, mastitis, displaced abomasum, and dyspepsia, with an average rectal temperature of +38.8 °C and a rumen motility five to six times per three minutes.

The control group (CG) (n = 25) included cows who showed no clinical symptoms of illness after calving and had all of their blood BHB measurements below 1.2 mmol/L in the 30-day period following calving. This group of cows had an average milk F/P of 1.2 or lower. The two groups of cows were housed together in a straw bedded calving pen. 

### 2.4. Measurements

Rumination, eating, and activity parameters were registered by RumiWatch noseband sensor (RWS) (ITIN + HOCH GmbH. Fütterungstechnik, Liestal, Switzerland). 

The RWS is made up of a liquid-filled pressure tube and a noseband halter with a pressure detector built in. The system algorithms classify items by recognizing unambiguous pressure peak clusters produced by jaw motions, which are then classified based on their behavioral traits [10]. 

Rumination, eating, and locomotion behaviors were registered by RWS. These data allowed the calculations of various other parameters, such as rumination time (time spent on ruminating chews, including chewing breaks of up to 5 s); eating time (time spent chewing food, including breaks of up to 5 s); drinking time (time spent drinking, including delays between gulps of up to 5 s); rumination chews (molars chewing during rumination for mechanical reduction of regurgitated materials into smaller bits); eating chews (total number of trepidation bites and mastication chews made when eating); drinking gulps (total amount of gulps taken while drinking); bolus (total amount of gulps taken while drinking); activity (sum of the duration of all walking bouts presented as minutes within a given recording period); up time (time spent feeding with the head positioned upwards (min/h)); down time (time spent feeding with the head positioned downwards (min/h)); and activity change (number of times switched between activities (between other activity, ruminating, eating, and drinking), as described by Zehner et al. 2017 [10]. 

Plasma ketone body levels (BHB) were determined on a capillary blood sample taken from the ear every day at 09:00 during clinical examination by using the Medi Sense and Free Style Optium H systems (Abbott, UK).

### 2.5. Duration of Measurements

All the cows were equipped with RWS from 40 days before calving allowing an adaptation period of 10 days. Rumination and eating parameters were registered by RWS starting from 30 days before calving and finished 30 days after calving. All cows had an ease calving evaluated at 1–2 points (on a five-point scale) since no assistance was required at the calving. 

### 2.6. Statistic and Software

Statistical data processing was carried out in the SPSS program for Windows (version 26.0; IBM Corp. Armonk, New York, NY) with normally distributed variables (according to the Kolmogorov–Smirnov test and graphical methods of histograms and Q-Q plots) of rumination behavior in cows. 

Considering the design of the study, we divided the data of cows into three stages for statistical evaluation: (1) the last thirty days before calving (from day −30 to −1 of the study); (2) day of calving (this day was excluded due to the possible influence of the calving process); (3) the first thirty days after calving (from day 1 to 30 of the study).

The significance of the differences between the mean values of two independent groups was assessed using the independent samples *t*-test. Student’s test for dependent samples was used to compare sample means from two periods of the experiment for the same groups.

Correlation analysis of indicators of ruminant behavior of individual cows before and after calving was carried out with the calculation of the Pearson correlation.

## 3. Results

### 3.1. Rumination and Activity Behavior Parameters before Calving

According to the statistical analysis (Table 3), differences between parameters recorded before calving were significant between groups, with the exception of rumination chews. The mean values in the CG group were from 3.25% (eating time 1, *p* < 0.05) to 19.07% (chews per bolus, *p* < 0.01) higher than in the TG group (of the following indicators: rumination time, eating time 1, drinking time, drinking gulps, bolus, chews per min, chews per bolus, downtime, maximal temperature, and activity change). The mean values of the TG group were higher than those of the CG cows for the following indicators: eating time 2 (19.55 %, *p* < 0.01), activity (6.63%, *p* < 0.01), uptime (3.46%, *p* < 0.05), average temperature (105.55%, *p* < 0.001), minimal temperature (7.00%, *p* < 0.051), other chews (110.47%, *p* < 0.001), eating chews 1 (23.72%, *p* < 0.01), and eating chews 2 (8.06%, *p* < 0.05).

### 3.2. Rumination and Activity Behavior Parameters on the Calving Day

On the day of calving, all the rumination and activity behavior parameters were significantly different between the groups (Table 4). The mean values of the time of rumination (9.53%, *p* < 0.01), eating time 1 and eating time (5.12%, *p* < 0.05 and 14.33 %, *p* < 0.01, respectively), bolus (24.79%, *p* < 0.001), chews per min (7.99%, *p* < 0.01), chews per bolus (34.62%, *p* < 0.001), uptime (8.72%, *p* < 0.01), downtime (23.94%, *p* < 0.001), minimal temperature (11.68%, *p* < 0.001), other chews (10.77%, *p* < 0.001), and eating chews 2 (24.16%, *p* < 0.001) were higher in the CG group than those of the TG group. Conversely, the mean values of the TG group were higher than those of the CG group in the following: drinking time (10.83%, *p* < 0.01), drinking gulps (4.97%, *p* < 0.05), activity (12.50%, *p* < 0.01), average temperature (96.67%, *p* < 0.001), maximal temperature (73.22%, *p* < 0.001), activity change (110.53%, *p* < 0.001), rumination chews (20.04%, *p* < 0.01), and eating chews 1 (4.13%, *p* < 0.05). 

### 3.3. Rumination and Activity Behavior Parameters after Calving

During the first thirty days after calving (Table 5), the average values of all rumination and activity behavior parameters differed significantly between groups, with the exception of chews per min. Higher mean values in the CG group compared to the TG group were determined by the following indicators: rumination time (6.88%, *p* < 0.05), eating time 1 (12.83%, *p* < 0.01), eating time 2 (21.43%, *p* < 0.001), bolus (8.64%, *p* < 0.01), chews per bolus (11.86%, *p* < 0.01), uptime (35.55%, *p* < 0.001), downtime (6.63%, *p* < 0.05), minimal temperature (12.33%, *p* < 0.01), maximal temperature (3.53%, *p* < 0.05), rumination chews (4.78%, *p* < 0.05), and eating chews 1 (10.84%, *p* < 0.01). We found lower mean values in the CG group (compared to cows in the TG group) for the following: drinking gulps, drinking time, activity and activity change (22.32–23.77%, *p* < 0.001), average temperature (132.32%, *p* < 0.001), other chews (34.79%, *p* < 0.001), and eating chews 2 (53.71%, *p* < 0.001).

After comparing parameters of rumination and activity behavior in the last 30 days before calving and on the day of calving, we found the largest increase in the TG group for these indicators: downtime, chews per bolus, maximum temperature, and rumination time (42.35–49.57%, *p* < 0.001) and the largest decrease for drinking gulps, time eating 2, eating chews 2, and activity (37.43–65.89%, *p* < 0.001). In the last 30 days before calving, the CG cows showed the largest increases in downtime, chews per bolus, other chews, and rumination time (46.91–87.17%, *p* < 0.001); the greatest decrease was in drinking gulps, change in activity, activity, and maximum temperature compared to TG cows (32.32–70.65%, *p* < 0.01) (Figure 1). 

Comparing the mean values of the TG group during the first 30 days post-calving with the last 30 days pre-calving (Figure 2), we determined that after calving, eating chews 2 of these cows increased the most (29.97%, *p* < 0.001) but eating chews 1 decreased (21.05%, *p* < 0.001), while the change in the readings of group TG were in the opposite direction for rumination chews, other chews, activity change, temperature (average, minimum and maximum), uptime, activity, bolus, drinking gulps and time, and eating time 2. After calving in the TG group of cows, the following indicators were decreased: uptime, other chews, eating chews 1, and eating time 1 and 2 (17.81–34.16 %, *p* < 0.01–0.001). In the CG group, such a sharp decrease in indicators was not found, only drinking gulps showed a reduction (21.77%, *p* < 0.001).

### 3.4. Rumination and Activity Behavior Parameters Correlation before and Alter Calving

In the CG group, all the rumination and activity behavior parameters in the periods before and after calving correlated positively, with the exception of changes in activity, drinking gulps, and maximum temperature (in the TG group—activity change, maximum temperature, uptime, chews per bolus, and chews per min also negatively correlated). The highest positive correlation coefficients between periods for the CG group were determined for the following studied indicators: rumination time, eating time 1, minimal temperature, chews per bolus, bolus, activity, eating time 2, and rumination chews (r = 0.412–0.771, *p* < 0.01). The mentioned correlation coefficients of the TG group were lower, but their highest values were also determined for the indicators of rumination time (r = 0.630, *p* < 0.01) and eating time 1 (r = 0.662, *p* < 0.01) (Figure 3).

## 4. Discussion

The automatic monitoring of intake and rumination demonstrated real potential for detecting health concerns following calving [9]. Significant reductions in rumination were also documented in dairy cows five to six days before the diagnosis of subclinical ketosis. This early identification of cows with health issues presents both benefits and challenges [14]. The RumiWatch technology can anticipate the start of calving in cattle three hours prior to the event [15]. In dairy farms, automated health-monitoring devices that detect activity and rumination can be used for identifying cows with digestive and metabolic issues [14]. To recognize cows with health problems in the early post-partum period, a comprehensive health-monitoring program is required [14]. Early detection of a health condition, before the emergence of noticeable symptoms, can benefit cows as it may lead to improved treatment outcomes and reduce the negative long-term impacts of the disease on the cow’s overall health and performance. [16]. Automatic rumination and feeding behavior monitoring in transition dairy cows shows promise for detecting health issues with the combination of both behaviors resulting in higher detection rates of post-partum diseases. Individual rumination behavior can be monitored more easily than individual feed intake with the technology employed in our study, making it more feasible and hence more likely to be applied on commercial farms. Rumination time, alone or in combination with other variables, has been studied as a technique for identifying parturition and sickness in commercial dairy cows [2,17]. Several studies have also been conducted to determine whether rumination activity is consistently lower in sick cows compared to healthy cows [17]. Previous research has linked rumination duration to clinical and subclinical health issues [14,18,19]. Cows with SCK were shown to have less rumination time than healthy cows in the first week after calving [18]. Rumination behavior may be a useful predictor of metabolic conditions [19], especially during the post-partum period, because it is likely to be influenced by changes in eating behavior [20]. 

According to Kaufman et al. [11], rumination monitoring throughout the transition phase may aid in the identification of multiparous cows at risk of developing SCK or suffering from SCK in conjunction with other health concerns. To utilize rumination data to help identify multiparous cows at risk of developing subclinical ketosis after calving, rumination should be monitored during the dry season to establish a baseline for each cow. Kaufman et al. [11] discovered that multiparous SCK cows had shorter rumination durations throughout the pre- and post-partum periods than healthy cows. Previous studies demonstrated that cows with a shorter rumination time had a higher incidence of clinical illness, such as mastitis, lameness, subclinical ketosis, and abomasal displacement [18]. Low dry matter intake (DMI) and short feeding times are also considered risk factors for SCK.

Rumination behavior could be a good biomarker for tracking metabolic disorders associated with a drop in DMI [18]. When aggregated across experimental circumstances, the amount of time dairy cattle spend eating varies greatly. White et al. [21] found a mean eating time of 284 min per day, ranging from 141 to 507 min per day. Some of the heterogeneity may be attributable to somewhat different criteria used to determine eating time among studies, but eating time is also heavily influenced by feed management, DMI, physical and chemical makeup of the diet, and inherent diversity across animals [21]. Ruminating and eating time appear to have a compensating relationship. According to Dado and Allen [22], the correlation coefficient between eating time and ruminating time for dairy cows with unlimited feed access was 0.62, showing that cows who spend less time eating ruminate longer. Predicting the amount of time cows spend chewing or ruminating can be a useful management tool for improving cow health, but the accuracy can be low due to the many interacting components [23].

According to King et al. [24], time spent ruminating decreased by 45 min per day eight days before a diagnosis of abomasal displacement, 25 min per day six days before subclinical ketosis, and 50 min per day five days before pneumonia. Kaufman et al. [11] discovered that there was no difference in rumination time between primiparous cows and the incidence of SCK, presumably due to the small number of primiparous cows with subclinical ketosis. Schirmann et al. [5] reported that cows with SCK and SCK and metritis had lower pre-partum DMI and continued to eat less for two to three weeks post-partum compared to the control group, but only cows with SCK pre-calving had decreased rumination. Paudyal et al. [25] discovered that ruminating time was reduced in both pre- and post-partum cows with SCK [25]. Paudyal et al. [25] predicted that drinking time would grow post-partum, reaching a maximum of minutes per day in early lactation, whereas our findings revealed a marked decrease (eight minutes per day in early lactation). However, drinking duration must be carefully considered as shorter, larger, or more frequent gulps may result in more water consumption. Furthermore, several factors can influence water intake, including changes in ambient temperature, increased water loss due to increased milk production, amount of feed consumed, salt and potassium consumption, diet dry matter, and physiological conditions and diseases [26]. In a previous study, we found that cows with SCK had shorter ruminating times and chews, as well as shorter drinking times, chews per minute, boluses, and chews per bolus. Seventeen days before the diagnosis of SCK, the healthy group spent more time ruminating and drinking than SCK. A decrease in rumination time was associated with a significant rise in the likelihood of SCK risk. Based on our findings, we can conclude that changes in rumination prior to the appearance of SCK could be used to aid in the early detection of the disease [27]. 

The goal of this study was to evaluate whether there were any rumination and activity changes in cows with subclinical ketosis around the time of calving. The results showed that the mean values of the healthy cows were higher from 3.25% (eating time 1, *p* < 0.05) to 19.07% (chews per bolus, *p* < 0.01) than in the SCK group for the following indicators: rumination time, eating time 1, drinking time, drinking gulps, bolus, chews per min, chews per bolus, downtime, maximal temperature, and activity change. Calving is a key time for both dairy cow and calf due to regrouping, nutrition changes, parturition, and the start of nursing [28,29,30]. Restless behavior is highlighted in the final two hours before calving and may be caused by discomfort [31]. Since primiparous and multiparous cows behaved differently, two models were employed to estimate calving time in each group using data from noseband sensors and 3D accelerometers. In both groups, lying bouts increased but rumination chews (RC) declined; additionally, boluses (B) fell while other activities increased considerably in multiparous and primiparous cows, respectively [15]. Under farm conditions, the RumiWatch system is an effective tool for predicting calving time [15]. 

We found that there are some changes in rumination behavior parameters on the calving day. The mean values of the time of rumination (9.53%, *p* < 0.01), eating time 1 and eating time (5.12%, *p* < 0.05 and 14.33 %, *p* < 0.01, respectively), bolus (24.79%, *p* < 0.001), chews per min (7.99%, *p* < 0.01), chews per bolus (34.62%, *p* < 0.001), uptime (8.72%, *p* < 0.01), downtime (23.94%, *p* < 0.001), minimal temperature (11.68%, *p* < 0.001), other chews (10.77%, *p* < 0.001), and eating chews 2 (24.16%, *p* < 0.001) were higher in the healthy group than those of the SCK group. In addition, after calving we found higher mean values in the healthy group compared to the SCK group, which were determined by the following indicators: rumination time (6.88%, *p* < 0.05), eating time 1 (12.83%, *p* < 0.01), eating time 2 (21.43%, *p* < 0.001), bolus (8.64%, *p* < 0.01), chews per bolus (11.86%, *p* < 0.01), uptime (35.55%, *p* < 0.001), downtime (6.63%, *p* < 0.05), minimal temperature (12.33%, *p* < 0.01), maximal temperature (3.53%, *p* < 0.05), rumination chews (4.78%, *p* < 0.05), and eating chews 1 (10.84%, *p* < 0.01). Comparing the mean values of the SCK group during the first 30 days post-calving with the last 30 days pre-calving, we determined that after calving, eating chews 2 of these cows increased the most (29.97 %, *p* < 0.001) but eating chews 1 decreased (21.05%, *p* < 0.001), while the change in the group SCK these readings were in the opposite direction as rumination chews, other chews, activity change, temperature (average, minimum and maximum), uptime, activity, bolus, drinking gulps and time, and eating time 2. After calving in the SCK group of cows, the following indicators most clearly decreased: uptime, other chews, eating chews 1, and eating time 1 and 2 (17.81–34.16%, *p* < 0.01–0.001). In the healthy group, such a sharp decrease in indicators was not found; the drinking gulps of these cows decreased the most (21.77%, *p* < 0.001). Previous research has found that ruminating and eating behavior is directly associated to clinical and subclinical health problems [19]. Many factors influence rumination and eating behavior including acute stress [32], disease [33]. The automatic monitoring of intake and rumination demonstrated some potential for detecting health concerns following calving [5]. In a previous study, we found that cows with subclinical ketosis had lower average values for the following parameters: rumination time and rumination chews (1.48 and 1.68 times, respectively, *p* < 0.001), drinking time (1.50 times; *p* < 0.001), chews per minute, bolus, and chews per bolus (1.12. 1.45. and 1.51 times, respectively, *p* 0.001). From the 15th day before the diagnosis of SCK, rumination time in the healthy group increased more than in the subclinical ketosis cows, specifically from 0.96% (17 day) to 187.79% (0 days *p* 0.001). Eating time in healthy cows was 48.92% higher at the start of the experiment and 91.97% at the end of the period when compared to subclinical ketosis (*p* < 0.001) [12]. Cows with SCK were shown to have a reduced rumination time compared to healthy cows in the first week after calving [19]. Rumination behavior may be a useful predictor of metabolic conditions [18], especially during the post-partum period because it is likely to be influenced by changes in eating behavior [34]. According to Kaufman et al. [11], rumination monitoring throughout the transition phase may aid in the identification of multiparous cows at risk of developing SCK or suffering from SCK in conjunction with other health concerns. To use rumination data to help identify multiparous cows at risk of developing subclinical ketosis after calving, rumination should be monitored during the dry season to establish a baseline for each cow [11].

The current study limitation was the lack of literature on physiological ranges for most of the feeding and ruminating parameters. We recommend greater research into physiological ranges with clinically healthy cows. Furthermore, more research is required to investigate the calving impact of rumination and activity behavior parameters.

## 5. Conclusions

According to the hypotheses of our study, we found that there are some changes in rumination behavior parameters around calving in cows with subclinical ketosis diagnosed within the first 30 days of calving. 

We found that before calving, those cows with a higher risk of having SCK diagnosed after calving had lower rumination time, eating time, drinking gulps, bolus, chews per min, chews per bolus, downtime, maximal temperature, and activity change. On the calving day, in cows with higher risk of SCK after calving, we found lower rumination time, eating time, chews per min, chews per bolus, uptime, downtime, minimal temperature, other chews, eating chews, drinking time, drinking gulps, activity, average temperature, maximal temperature, activity change, rumination chews, and eating chews. After calving in cows with SCK, we found lower rumination time, eating time 1, eating time 2, bolus, chews per bolus, uptime, downtime, minimal temperature, maximal temperature, rumination chews, and eating chews. In this period, we found higher drinking gulps, drinking time, activity, activity change, average temperature, other chews, and eating chews in cows with SCK. 

Based on this, we can conclude that by tracking changes in rumination behavior parameters registered with RWS (such as rumination time, eating time, drinking time, drinking gulps, bolus, chews per minute, chews per bolus, downtime, maximal temperature, and activity change) before, during, and after calving, we can identify cows with a higher risk of SCK. 

## Figures and Tables

**Figure 1 animals-13-00595-f001:**
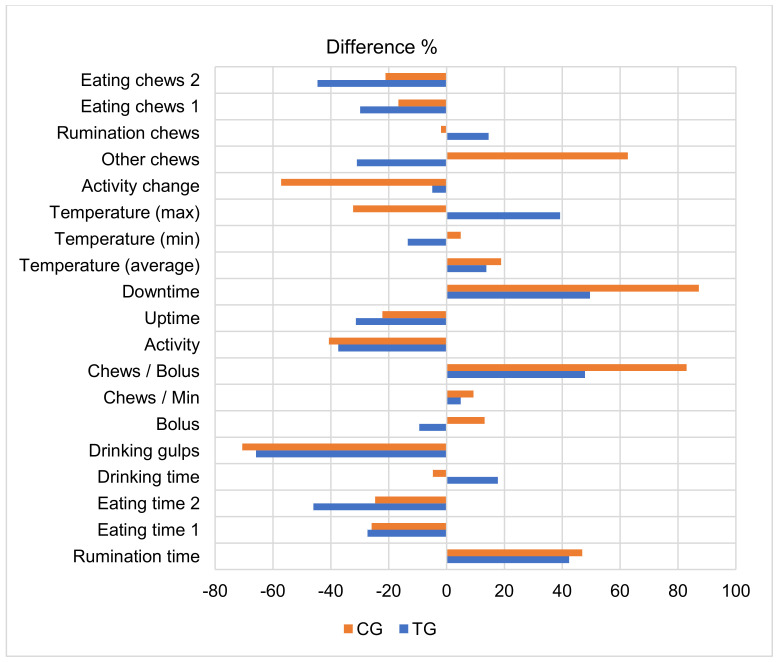
Comparison of the rumination behavior parameters during the last 30 days before calving and on the calving day. The differences between the mean values of the periods (period: the last 30 days before calving = 100%) are statistically significant. TG—subclinical ketosis group; CG—control group.

**Figure 2 animals-13-00595-f002:**
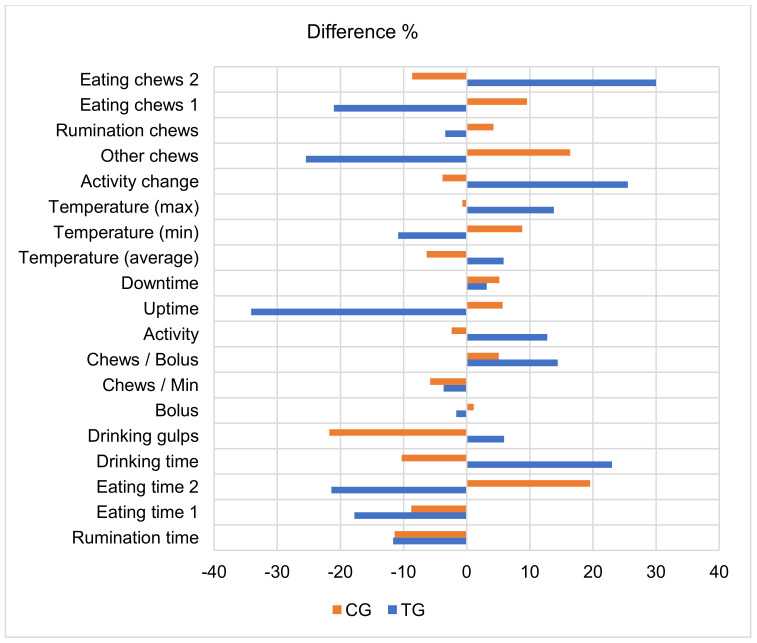
Comparison of parameters of rumination behavior during the last 30 days before calving and the first 30 days after calving. The differences between the mean values of the periods (period: the last 30 days before calving = 100 %) are statistically significant. TG—subclinical ketosis group; CG—control group.

**Figure 3 animals-13-00595-f003:**
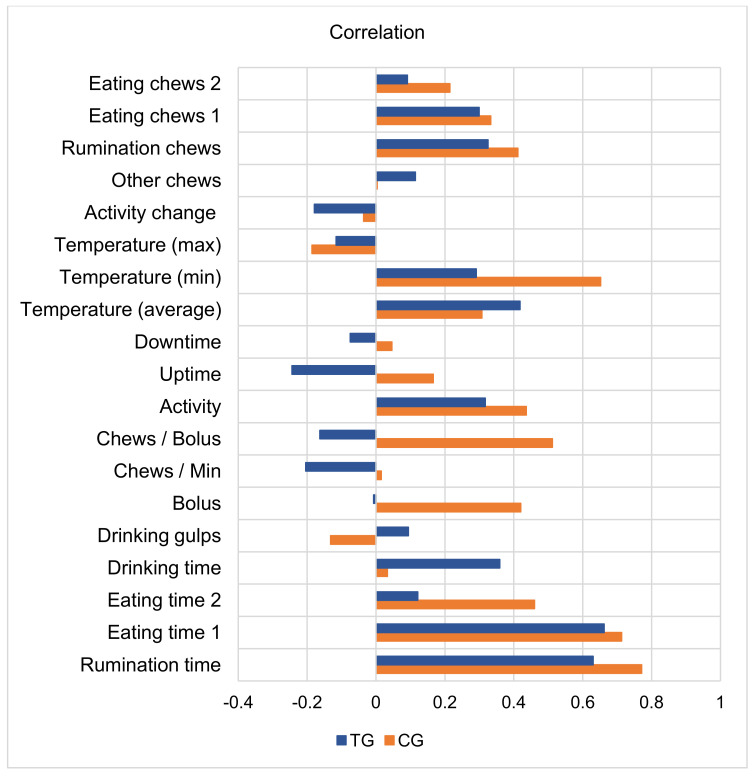
Correlation of the rumination behavior indicators before and after calving by groups. Correlations of indicators of individual cows between the periods before calving and after calving are statistically significant. TG—subclinical ketosis group, CG—control group.

**Table 1 animals-13-00595-t001:** Components of the total mix ration (TMR).

Feed	Fresh Dairy Cows	Dry Cows (40 Days before Calving)
Barley grain, 74% DM (kg)	3	0
Corn grain 56% DM (kg)	3	0
Rapeseed meal 36% protein (kg)	2	1.2
Soy meal is 46% protein (kg)	1	0
Beetroot molasses (kg)	0.5	0
Grass silage 27% DM (kg)	20	8
Maize silage 27% DM (kg)	24.5	1.2
Wheat straws (kg)	0.6	7.5
Nordic Fat (Bergafat) 300 (kg)	0.24	0
Water (kg)	5	4
Grain mixture (kg)	5	
Mineral–vitamin supplement for dairy cows (kg)	0.30	0
Mineral–vitamin supplement for dry cows (kg)	0	0.15

**Table 2 animals-13-00595-t002:** Chemical composition of feeding rations for dry and fresh dairy cows.

Parameters	Fresh Dairy Cows	Dry Cows
Dry matter (DM) (%)	45.0	46.0
Dry matter intake (kg DM/d)	27.5	12.1
Net energy lactation (MJ/kg DM)	6.42	4.38
Crude protein (g/kg DM)	172	108
Crude Fat (g/kg DM)	47	25
Fatty acids (g/kg DM)	34	9
Protein balance in rumen (g/kg DM)	23	10
Neutral detergent fiber (g/kg DM)	291	629
Starch (g/kg DM)	205	25
Acid detergent fiber (ADF) (g/kg DM)	180	170
Acid detergent lignin (ADL) (g/kg DM)	20	18
Sugar (g/kg DM)	62	25

**Table 3 animals-13-00595-t003:** RumiWatch noseband (RWS) parameters before calving (from day “−30” to day “−1”) by groups.

RumiWatch Noseband Parameters	CG		TG	
Rumination time	24.28	±1.165	22.67 *	±1.088
Eating time 1	9.23	±0.443	8.93 *	±0.429
Eating time 2	4.45	±0.214	5.32 **	±0.255
Drinking time	1.26	±0.060	1.13 **	±0.044
Drinking gulps	228.25	±10.956	206.21 **	±9.8908
Bolus	25.86	±1.231	24.29 *	±1.126
Chews/min	70.58	±3.308	67.64 *	±3.247
Chews/bolus	14.21	±0.612	11.5 **	±0.512
Activity	53.96	±2.550	57.54 **	±2.739
Uptime	28.32	±1.319	29.3 *	±1.401
Downtime	23.23	±1.112	22.11 *	±1.041
Temperature (average)	6.31	±0.301	12.97 ***	±0.603
Temperature (min)	33.56	±1.601	35.94 *	±1.714
Temperature (max)	33.38	±1.601	28.1 **	±1.329
Activity change	8.88	±0.426	8.42 *	±0.404
Other chews	109.54	±5.258	230.55 ***	±11.066
Rumination chews	1270.9	±61.001	1306.3	±62.701
Eating chews 1	413.8	±19.862	511.94 **	±24.543
Eating chews 2	239.59	±11.500	258.89 *	±12.407

The differences between the mean values of the groups are statistically significant: *—*p* < 0.05; **—*p* < 0.01; and ***—*p* < 0.001. TG—subclinical ketosis group; CG—control group.

**Table 4 animals-13-00595-t004:** RumiWatch noseband (RWS) parameters on the calving day.

RumiWatch Noseband Parameters	CG		TG	
Rumination time	35.67	±1.712	32.27 **	±1.549
Eating time 1	6.84	±0.328	6.49 *	±0.314
Eating time 2	3.35	±0.165	2.87 **	±0.138
Drinking time	1.20	±0.058	1.33 **	±0.064
Drinking gulps	67.00	±3.216	70.33 *	±3.374
Bolus	29.25	±1.406	22.00 ***	±1.056
Chews/min	77.13	±3.706	70.97 **	±3.404
Chews/bolus	26.00	±1.248	17.00 ***	±0.816
Activity	32.00	±1.536	36.00 **	±1.728
Uptime	22.03	±1.057	20.11 **	±0.965
Downtime	43.48	±2.087	33.07 ***	±1.587
Temperature (average)	7.50	±0.360	14.75 ***	±0.708
Temperature (min)	35.20	±1.690	31.09 **	±1.492
Temperature (max)	22.59	±1.084	39.13 ***	±1.878
Activity change	3.80	±0.182	8.00 **	±0.384
Other chews	178.20	±8.554	159.02 **	±7.632
Rumination chews	1246.30	±59.820	1496.03 **	±71.808
Eating chews 1	344.75	±16.548	359.01 *	±17.232
Eating chews 2	189.00	±9.072	143.33 **	±6.880

The differences between the mean values of the groups are statistically significant: *—*p* < 0.05; **—*p* < 0.01; and ***—*p* < 0.001. TG—subclinical ketosis group; CG—control group.

**Table 5 animals-13-00595-t005:** RumiWatch noseband (RWS) indicators after calving (from day “1” to day “30”).

RumiWatch Noseband Indicator	CG		TG	
Rumination time	21.51	±1.035	20.03 *	±0.967
Eating time 1	8.42	±0.404	7.34 **	±0.352
Eating time 2	5.32	±0.253	4.18 ***	±0.209
Drinking time	1.13	±0.054	1.39 ***	±0.067
Drinking gulps	178.57	±8.576	218.43 ***	±10.485
Bolus	26.15	±1.255	23.89 **	±1.147
Chews/min	66.49	±3.195	65.16	±3.128
Chews/bolus	14.93	±0.717	13.16 **	±0.632
Activity	52.67	±2.528	64.89 ***	±3.115
Uptime	29.93	±1.437	19.29 ***	±0.926
Downtime	24.43	±1.173	22.81 *	±1.095
Temperature (average)	5.91	±0.284	13.73 ***	±0.659
Temperature (min)	36.51	±1.752	32.01 **	±1.536
Temperature (max)	33.15	±1.591	31.98 *	±1.535
Activity change	8.54	±0.410	10.57 ***	±0.507
Other chews	127.49	±6.124	171.84 ***	±8.248
Rumination chews	1325	±63.601	1261.7 *	±60.563
Eating chews 1	453.29	±21.758	404.16 **	±19.400
Eating chews 2	218.9	±10.507	336.47 ***	±16.156

The differences between the mean values of the groups are statistically significant: *—*p* < 0.05; **—*p* < 0.01; and ***—*p* < 0.001. TG—subclinical ketosis group; CG—control group.

## Data Availability

The data presented in this study are available within the article.

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
