# Peer review of "Change in Rumination Behavior Parameters around Calving in Cows with Subclinical Ketosis Diagnosed during 30 Days after Calving"

_animals, 2023, doi:10.3390/ani13040595_

Round 1
Reviewer 1 Report
The manuscript describes a straight forward investigation of rumination behavior in cows with subclinical ketosis diagnosed. The paper describes and presents good quality data to identify cows with a higher risk of SCK by tracking changes in rumination behavior parameters registered with RWS.
Given using sensors for health detection is an exciting and active field, the introduction seems to be a little basic and could have included a number of studies relating to RumiWatch to broaden the scope of the findings presented.
Similarly, the discussion could have considered the role of the RumiWatch noseband sensor in conjunction with other devices, such as MooMonitor+. Moreover, whether these data are valuable for cows with SCK and one or more other health problems.
The English should be improved.
Author Response
Dear Reviewer,
Authors are very thankful for the comments, which help us to improve the manuscript. All changes proposed have been included in the manuscript and highlighted in yellow and track changes.
Best Regards,
Prof. Ramunas Antanaitis
|
Question |
Answers |
|
Given using sensors for health detection is an exciting and active field, the introduction seems to be a little basic and could have included a number of studies relating to RumiWatch to broaden the scope of the findings presented. Similarly, the discussion could have considered the role of the RumiWatch noseband sensor in conjunction with other devices, such as MooMonitor+. Moreover, whether these data are valuable for cows with SCK and one or more other health problems. |
Corrected in whole manuscript. |
|
The English should be improved. |
corrected by a native speaker
|
Reviewer 2 Report
Review, paper no. animals-2173917 entitle „Change In Rumination Behavior Parameters Around Calving In Cows With Subclinical Ketosis Diagnosed During 30 Days After Calving”. This is an important area for improving cattle production. This is a well-organized study, with sufficient methodology and adequate description of the results. Authors' research has shown interesting relationships. The manuscript idea is somehow new with great interest and it is well written as well. The authors have used the standard journal format in manuscript writing. The manuscript contains several inaccuracies in methodology. Generally, the article has some interesting findings which could be worth publication.
Specific comments:
Overall it is still quite well written although the „Statistical analysis” could be more clearer.
Simple Summary: Correct.
Abstract: Is sufficiently presented (methods, results, general conclusions).
Line 40-43. clearly indicate which parameters in which direction changes will be the prediction of SCK. Clearly indicate practical usefulness.
Introduction: The introduction section is sufficient and analytically and adequately covers the need for the study.
Line 105 „average feed intake of 19 kg DM/day”. Lactating cows?
Materials and Methods: The methodology is sufficiently presented. However, it has a few inaccuracies.
Table 1. Water (kg) ? Revise for clarity.
Line 122. beta-hydroxybutyrate acid
Line 122. What method was used to measure BHB. How long after feeding.
Results:
Table 3. You must present data that it was clinical ketosis. We don't know from work.
Discussion:
Please explain whether your own research concerns clinical or subclinical ketosis. Revise (results and discussion).
Could authors define possible limitations of the study?
Conclusion:
Clearly indicate the direction of changes in the tested parameters. Consider the conclusion of your own research. Conclusions must be clear.
Author Response
Dear Reviewer,
Authors are very thankful for the comments, which help us to improve the manuscript. All changes proposed have been included in the manuscript and highlighted in yellow and track changes.
Best Regards,
Prof. Ramunas Antanaitis
|
Question |
Answers |
|
Line 40-43. clearly indicate which parameters in which direction changes will be the prediction of SCK. Clearly indicate practical usefulness. |
We added information – “We found that before calving, these cows with a higher risk of having SCK diagnosed after calving had lower rumination time, eating time, drinking gulps, bolus, chews per min, chews per bolus, downtime, maximal temperature, and activity change. On the calving day, in cows with higher risk of SCK after calving, we found lower rumination time, eating time, chews per min, chews per bolus, uptime, downtime, minimal temperature, other chews, eating chews, drinking time, drinking gulps, activity, average temperature, maximal temperature, activity change, rumination chews, and eating chews. After calving in cows with SCK, we found lower rumination time, eating time 1, eating time 2, bolus, chews per Bolus, uptime, downtime, minimal temperature, maximal temperature, rumination chews, and eating chews. In this period we found higher drinking gulps, drinking time, activity, activity change, average temperature, other chews and eating chews in cows with SCK. From a practical point of view, we can recommend that by tracking changes in rumination behavior parameters registered with RWS (such as rumination time, eating time, drinking time, drinking gulps, bolus, chews per minute, chews per bolus, downtime, maximal temperature, and activity change) before, during, and after calving, we can identify cows with a higher risk of SCK in the herd”
|
|
Line 105 „average feed intake of 19 kg DM/day”. Lactating cows? |
Corrected to –“… intake of 23.2…” |
|
Materials and Methods: The methodology is sufficiently presented. However, it has a few inaccuracies. |
Corrected according your recommendation |
|
Table 1. Water (kg) ? Revise for clarity. |
Revised |
|
Line 122. beta-hydroxybutyrate acid |
Corrected to – “beta-hydroxybutyrate acid” |
|
Line 122. What method was used to measure BHB. How long after feeding. |
We added information – “Plasma ketone body levels were determined every day at 09:00 am. during clinical examination by using the Medi Sense and Free Style Optium H systems (Abbott, Great Britain) and a capillary blood sample taken at the ear” |
|
Table 3. You must present data that it was clinical ketosis. We don't know from work. |
We corrected to – “TG - subclinical ketosis group” |
|
Discussion: Please explain whether your own research concerns clinical or subclinical ketosis. Revise (results and discussion).
|
We are sorry for the mistake. It may be subclinical ketosis (SCK). We corrected the whole manuscript
|
|
Could authors define possible limitations of the study? |
We added information – “The current study limitation was the lack of literature on physiological ranges for most of the feeding and ruminating parameters could be found. We recommend greater research into physiological ranges with clinically healthy cows based on this. Furthermore, more research is required to investigate the calving impact of the investigated parameters”
|
|
Conclusion: Clearly indicate the direction of changes in the tested parameters. Consider the conclusion of your own research. Conclusions must be clear.
|
We added information in conclusion section – “We found that before calving, these cows with a higher risk of having SCK diagnosed after calving had lower rumination time, eating time, drinking gulps, bolus, chews per min, chews per bolus, downtime, maximal temperature, and activity change. On the calving day, in cows with higher risk of SCK after calving, we found lower rumination time, eating time, chews per min, chews per bolus, uptime, downtime, minimal temperature, other chews, eating chews, drinking time, drinking gulps, activity, average temperature, maximal temperature, activity change, rumination chews, and eating chews. After calving in cows with SCK, we found lower rumination time, eating time 1, eating time 2, bolus, chews per Bolus, uptime, downtime, minimal temperature, maximal temperature, rumination chews, and eating chews. In this period we found higher drinking gulps, drinking time, activity, activity change, average temperature, other chews and eating chews in cows with SCK” |
Reviewer 3 Report
in attachments

Author Response
Dear Reviewer,
Authors are very thankful for the comments, which help us to improve the manuscript. All changes proposed have been included in the manuscript and highlighted in yellow and track changes.
Best Regards,
Prof. Ramunas Antanaitis
|
Question |
Answers |
|
Before starting the review, it is necessary to determine whether this work has already been published before or whether it is a salami publication, because the authors have so far published a large number of thematically and methodologically similar works. The works are properly cited and discussed in the context of this paper, which is the subject of the review. However, it is necessary for the authors together with the editors and reviewers to consider whether this latest work is actually just a part of some previous research or is to a greater extent new research. Salami publications reduce the reputation of publishers, so this should be discussed.
|
In this manuscript, we presented results from our project - "Identification of biomarkers from automatic health monitoring system for early diagnosis of diseases after calving and reproductive success" Project idea: research conducted by the leader and researcher group of the intended project are aimed at identifying the variation of certain biologic markers (biomarkers) in the cases of fresh cow diseases (milk fever, retained placenta, sub- clinical ketosis, sub- clinical ruminal acidosis, sub- clinical mastitis, sub –clinical metritis, abomasal displacement (left and right), lameness and others). Specifically, the identification is performed before disease manifestation, during onset of clinical symptoms. The diseases after calving, such as ketosis, abomasal displacement, mastitis, metritis, endometritis, acidosis, etc., have been selected as the research object due to the frequent manifestation and considerable economic damage caused by them to the farms. The project belongs to the field of early diagnosis of fresh dairy cow diseases and evaluation of reproduction success with reference to changes of certain markers. The markers which can be recorded continuously without causing any stress or discomfort to an animal, using modern innovative methods, have been selected for the research. These methods are automatic health monitoring systems, which consists of automatic milking system number 1 (AMS1), automatic milking system number 2 (AMS2) and rumen behavior system (RBS). Work aim: to assess the specificity of biomarkers measured by automated health monitoring system (AHMS) (automatic milkings and rumen behavior systems) for early diagnosis of fresh dairy cow’s disease and evaluation of reproduction success. Work objectives: 1. To determine if the biomarkers from automatic milking system (AMS1): milk yield, milking time, milk composition (fat, protein, somatic cell count), blood in milk, electrical milk conductivity, body weight and cow activity can serve as biomarkers of diseases after calving and diseases of the reproduction system. 2. To determine if the biomarkers from automatic milking system (AMS2): milk progesterone, milk lactatdehydrogenase, milk beta – hydroxybutyrate, cows body condition score can serve as biomarkers of diseases after calving and diseases of the reproduction system. 3. To determine if the biomarkers from rumen behavior system (RBS): reticulorumen ph, temperature and rumination parameters registered before and after calving can serve as biomarkers of diseases after calving and diseases of the reproduction system. 4. According to results of current study, create and test in practice new technology models from biomarkers form two different automatic milking systems (AMS1 and AMS2) and rumen behavior system (RBS) for early diagnosis of diseases after calving and evaluation of reproduction success. In the current manuscript, we answer task number 3 "To determine if the biomarkers from rumen behavior system (RBS): reticulorumen ph, temperature and rumination parameters registered before and after calving can serve as biomarkers of diseases after calving and diseases of the reproduction system" We used similar methodology in all experiments for which results were published, as confirmed by this project. However, the objectives, results, discussions, and conclusions of each paper are different.
|
|
When it comes to tables, they need to be visually and technically arranged according to MDPI standards, because obviously some row has been moved or some line is not well drawn. |
Corrected in whole manuscript |
|
Abbreviations below the tables do not correspond to the title. Below the table it says CG-control group and CK-clinical ketosis group. You were dealing with subclinical ketosis. I suggest that you do not write abbreviations in the header of the table. Instead of CG put Healthy group, and instead of CK put Subclinical ketosis group. |
We are sorry for the mistake. It may be subclinical ketosis (SCK). We corrected the whole manuscript
|